# Antagonistic Effects and Volatile Organic Compound Profiles of Rhizobacteria in the Biocontrol of *Phytophthora capsici*

**DOI:** 10.3390/plants13223224

**Published:** 2024-11-16

**Authors:** José Luis Ávila-Oviedo, Carlos Méndez-Inocencio, María Dolores Rodríguez-Torres, María Valentina Angoa-Pérez, Mauricio Nahuam Chávez-Avilés, Erika Karina Martínez-Mendoza, Ernesto Oregel-Zamudio, Edgar Villar-Luna

**Affiliations:** 1Instituto Politécnico Nacional, Centro Interdisciplinario de Investigación para el Desarrollo Integral Regional (CIIDIR), Unidad Michoacán, Justo Sierra 28, Col. Centro, Jiquilpan 59510, Michoacán, Mexico; 1355646g@umich.mx (J.L.Á.-O.); cmendezi@ipn.mx (C.M.-I.); mdrodriguezt@ipn.mx (M.D.R.-T.); vangoa@ipn.mx (M.V.A.-P.); ery_099@hotmail.com (E.K.M.-M.); 2Laboratorio de Bioquímica y Biología Molecular, División de Ingeniería Bioquímica, Tecnológico Nacional de México/ITS de Ciudad Hidalgo, Av. Ing. Carlos Rojas Gutiérrez No. 2120 Fracc. Valle de La Herradura C.P., Hidalgo 61100, Michoacán, Mexico; nchavez@cdhidalgo.tecnm.mx

**Keywords:** biological control, fungicide alternatives, microbial interactions, plant disease management, sustainable agriculture

## Abstract

*Phytophthora capsici* is a devastating pathogen in horticultural crops, particularly affecting *Capsicum annuum* (pepper). The overuse of chemical fungicides has led to resistance development, necessitating alternative strategies. This study investigates the antagonistic effects of four rhizobacterial isolates (*Bacillus* sp., *Pseudomonas putida*, *Bacillus subtilis*, *Bacillus amyloliquefaciens*) against *P. capsici*, focusing on the production of volatile organic compounds (VOCs). Using in vitro dual culture assays, we observed a significant inhibition of mycelial growth and sporangia production, especially by *B. subtilis* and *B. amyloliquefaciens*. The GC-MS/SPME-HS analysis identified key VOCs responsible for these antagonistic effects. Our findings demonstrate that specific rhizobacteria and their VOCs offer a promising biocontrol strategy, potentially reducing the reliance on chemical fungicides and contributing to sustainable agriculture.

## 1. Introduction

*Capsicum annuum* L. (Solanales: Solanaceae) is a globally significant horticultural crop, widely used in both the food industry and traditional medicine. Its fruits are a valuable source of essential micro- and macronutrients, including vitamins A, C, and E; minerals; and antioxidant compounds [1], which have demonstrated beneficial properties for human health, such as their potential to prevent and treat chronic degenerative diseases, including cancer and cardiovascular diseases [2]. Furthermore, *C. annuum* (peppers) represent one of the most economically important horticultural crops, being widely cultivated in various regions of the world, contributing both to the livelihoods of small-scale farmers and to the global economy.

However, *C. annuum* production is severely threatened by the incidence of diseases, with one of the most devastating caused by *Phytophthora capsici*. This oomycete induces a disease commonly known as pepper wilt, with symptoms including root and stem rot, as well as leaf and fruit blight [3]. Controlling this pathogen represents a significant challenge, particularly in horticultural crops, where effective disease management is critical for ensuring both the sustainability and profitability of production.

*P. capsici* is a cosmopolitan, hemibiotrophic pathogen that can infect various horticultural crops at any phenological stage, particularly those from the *Solanaceae* [4] and *Cucurbitaceae* [5] families. Infection begins with the production of reproductive structures, either sexual or asexual [6]. In the asexual cycle, sporangia are produced, which can either generate germ tubes that penetrate the host’s surface, or release biflagellate zoospores that swim through water to reach the host [7]. Once adhered, *P. capsici* penetrates the plant through natural openings, wounds, or directly by employing enzymes that facilitate its establishment and proliferation within the host tissues [5].

The management of *P. capsici* in horticultural crops has traditionally relied on the intensive use of synthetic fungicides such as metalaxyl, mefenoxam, cymoxanil, dimethomorph, fluazinam, fosetyl-A1, oligochitosan, oxathiapiprolin, zoxamide, and azoxystrobin [8]. However, *P. capsici* has developed resistance to several compounds [5,9], necessitating increased application rates and leading to negative environmental and human health impacts. In response, there has been a growing interest in developing more sustainable and environmentally friendly disease management strategies, such as the use of beneficial microorganisms, including plant growth-promoting bacteria (PGPB), which can produce volatile organic compounds (VOCs) and diffusible organic compounds with antimicrobial properties [10].

Various species of *Bacillus* have been reported to exhibit an antagonistic activity against plant pathogens through antibiosis, producing compounds such as benzaldehyde, nonanal, benzothiazole, and acetophenone [11]. Similarly, certain strains of *Pseudomonas* produce phenazine-1-carboxamide (PCN), DAPG (2,4-diacetylphloroglucinol), and phenazine-1-carboxylic acid (PCA), which have inhibitory effects against *Rhizoctonia solani* and *Seiridium cardinale*, respectively [12]. Other compounds, such as 2,4-DAPG, phenazine-1-carboxylic acid, methyl heptenone, d-limonene, and 2-undecanone, have been shown to affect the infective structures of various *Pythium* species [13,14].

It has been reported that PGPB produce distinct VOC profiles, both in terms of abundance and chemical composition, depending on the intra- or interspecific interactions they encounter, as well as the available nutrient sources [15,16]. Nevertheless, studies on the antagonistic activity of bacterial VOCs against oomycete pathogens, such as *P. capsici*, remain scarce. Therefore, this study aims to investigate the in vitro antagonistic effects of four bacterial isolates (*Bacillus* sp., *Pseudomonas putida*, *Bacillus subtilis*, and *Bacillus amyloliquefaciens*) against *P. capsici*, and to characterize the VOC profiles responsible for this antagonistic activity, with the goal of offering sustainable alternatives for disease management in horticultural crops. We hypothesize that specific rhizobacterial isolates produce VOCs that exhibit antagonistic effects against *P. capsici*, offering sustainable alternatives for disease management in horticultural crops.

## 2. Results

### 2.1. In Vitro Antagonistic Activity

In vitro confrontations between bacterial isolates and *Phytophthora capsici* showed differentiated effects on the pathogen’s mycelium after four days of incubation. In the control group, which consisted solely of *P. capsici*, the mycelium displayed a whitish coloration, typical of the oomycete. However, in the BMBH/Pc and BMBI/Pc treatments (exposed to *Bacillus* sp. and *Pseudomonas putida*, respectively), the mycelium exhibited a brownish hue (Figure 1). Additionally, in both treatments, *Bacillus* sp. and *P. putida* promoted the radial growth of *P. capsici* mycelium, showing significant differences compared to the control (one-way ANOVA, F = 206.63, df = 4, *p* ≤ 0.05).

On the other hand, some bacteria caused a reduction in mycelial density and pathogen growth. This was particularly evident in the treatments with *P. putida* (BMBI/Pc), *Bacillus amyloliquefaciens* (BMBC/Pc), and *Bacillus subtilis* (BMBA/Pc), with inhibition percentages of 12.62%, 37.11%, and 49.53%, respectively (one-way ANOVA, F = 206.63, df = 4, *p* ≤ 0.05). In these treatments (BMBC/Pc, BMBA/Pc, BMBH/Pc, and BMBI/Pc), a significant reduction in hyphal density and thinning was observed compared to the control (Figure 2, 10×). Furthermore, the treatments with *B. subtilis* and *B. amyloliquefaciens* induced hyphal shortening, breaking, and swelling. Bulbous hyphae were also visualized in the BMBA/Pc treatment (Figure 2, 40×). In the BMBH/Pc treatment, an increase in sporangia formation was observed, indicating an alteration in the reproductive structures (Figure 3).

### 2.2. Morphological Analysis and Sporangia Quantification

Sporangium production is crucial for the infective phase of *P. capsici*, so its production during its interaction with bacteria was evaluated. The results showed two main effects; on the one hand, exposure to *Bacillus* sp. led to a 28% increase in sporangium production compared to the control, while exposure to *B. amyloliquefaciens* and *B. subtilis* resulted in a reduction of 95% and 98%, respectively (one-way ANOVA, F = 10.19, df = 4, *p* ≤ 0.05) (Figure 3).

In addition to this, alterations in sporangium morphology were observed. Morphometric analysis revealed that *B. subtilis* was the only bacterium that significantly reduced the average sporangium length (24.85 µm) and width (24.23 µm) (Figure 4).

### 2.3. Severity Analysis

To determine whether the observed alterations in sporangia affected the infective capacity of *P. capsici*, a severity assay was performed on detached leaves of *Capsicum annuum* cv. California Wonder. At 8 days post-inoculation (dpi), it was found that mycelium from the BMBI/Pc, BMBH/Pc and BMBI/Pc treatments did not show a significant reduction in disease severity compared to the control (one-way ANOVA, F = 243.76, df = 5, *p* ≤ 0.05). In contrast, mycelium from the BMBC/Pc and BMBA/Pc treatments significantly reduced disease severity compared to the other treatments, including the control (one-way ANOVA, F = 243.76, df = 5, *p* ≤ 0.05). No disease symptoms were observed in the negative control (Figure 5).

### 2.4. Volatile Organic Compounds (VOCs) During Rhizobacteria-P. capsici Interaction

In the individual culture systems, a total of 32 volatile organic compounds (VOCs) were identified across *P. capsici* (Pc) and the four rhizobacterial isolates, *Bacillus* sp. (BMBH), *Bacillus subtilis* (BMBA), *Bacillus amyloliquefaciens* (BMBC), and *Pseudomonas putida* (BMBI) (Table 1). The VOC profiles varied significantly among these treatments. The control system (Pc alone) predominantly produced fluoroethylene (57.84 ± 2.31%), dimethyl ether (22.07 ± 0.66%), and cyclopropyl carbinol (9.23 ± 0.28%). Notably, isobutane was detected exclusively in BMBH (29.98 ± 1.20%) and BMBI (9.18 ± 0.37%), while it remained absent in Pc, BMBA, and BMBC. Similarly, 3-methyl-1-butanol and DL-alanine were present in BMBH (5.20 ± 0.21% and 4.33 ± 0.17%, respectively) and BMBI (6.45 ± 0.26% and 6.74 ± 0.27%, respectively), but they were not detected in the other treatments. BMBA and BMBC exhibited distinct VOC profiles; acetoin was significantly abundant in BMBA (25.91 ± 1.04%) and BMBC (35.51 ± 1.42%) but absent in Pc, BMBH, and BMBI. Ethyl methoxyacetate was uniquely present in BMBC at 19.14 ± 0.77%. These findings indicate that each rhizobacterial isolate possesses a unique VOC signature when cultured individually, suggesting specific metabolic pathways.

The VOC profiles in the dual culture systems, where each rhizobacterial isolate was co-cultured with *P. capsici* (BMBA/Pc, BMBH/Pc, BMBC/Pc, BMBI/Pc), showed notable alterations compared to the individual cultures (Table 2). In BMBA/Pc, acetoin was markedly abundant at 44.53 ± 1.80%, an increase from its level in BMBA alone. Ethyl methoxyacetate appeared in BMBA/Pc at a significant level (26.19 ± 1.00%), despite being undetected in the individual BMBA culture. BMBC/Pc exhibited high levels of L-lactic acid (32.78 ± 1.30%) and acetoin (20.06 ± 0.80%), indicating an enhanced production during interaction with *P. capsici*. In BMBH/Pc, isobutane remained abundant (28.04 ± 1.10%), and 2-nitroethanol was notably present at 27.88 ± 1.10%, which was not detected in BMBH alone. BMBI/Pc showed significant amounts of hydrazoic acid (19.57 ± 0.80%) and 2-nitroethanol (19.38 ± 0.80%), compounds that were either absent or present in lower abundance in the individual BMBI culture. These results suggest that the interaction between each rhizobacterial isolate and *P. capsici* leads to distinct VOC profiles, potentially due to altered metabolic pathways activated during the co-culture.

Comparing the VOC profiles of the individual and dual systems reveals significant differences in the metabolite production upon interaction with *P. capsici*. Acetoin, for example, increased substantially in BMBA/Pc (44.53 ± 1.80%) compared to BMBA alone (25.91 ± 1.04%), suggesting an upregulation of its biosynthesis during interaction. In contrast, BMBC showed a decrease in acetoin production in the dual system (20.06 ± 0.80%) compared to its individual culture (35.51 ± 1.42%). Ethyl methoxyacetate, absent in BMBA individually, was produced at a high relative abundance in BMBA/Pc (26.19 ± 1.00%). Some compounds present in individual cultures were reduced or not detected in the dual systems; for instance, fluoroethylene, prominent in Pc alone, was significantly reduced or absent in the dual cultures. These variations indicate that the rhizobacteria-*P. capsici* interactions influence VOC production, potentially affecting microbial communication and antagonistic activities. The enhanced or suppressed production of specific VOCs in the dual systems underscores the dynamic metabolic responses occurring during microbial interactions, which could be pivotal in understanding the mechanisms of biocontrol and pathogenesis.

The hierarchical clustering heatmap (Figure 6) provides a detailed visualization of the VOC profiles across the individual and dual systems, revealing distinct patterns of metabolite production influenced by the interaction between rhizobacteria and *P. capsici*. The VOCs were classified into two main groups based on their abundance patterns; Group 1 comprised VOCs that were produced in greater abundance in the dual systems compared to the individual systems (either bacteria or *P. capsici* alone), while Group 2 included VOCs identified in the individual bacterial treatments that were not produced during interaction with *P. capsici* in the dual systems.

In the BMBH/Pc system, compounds such as 3-methyl-1-butanol and DL-alanine were prominent in Group 1, indicating an enhanced production when *Bacillus* sp. interacted with *P. capsici*. Conversely, 2-methyl-1-butanol and 2-nitropropane were categorized into Group 2, as they were produced by *Bacillus* sp. alone but were absent in the dual system, suggesting that the interaction may suppress their synthesis. Similarly, in the BMBI/Pc system, isobutane and 2-methyl-1-butanol were abundant in Group 1, highlighting increased synthesis during the interaction between *Pseudomonas putida* and *P. capsici*. In contrast, compounds like L-lactic acid, 1-pentanal, and 2,2′,6,6′-tetra-tert-butyl-4,4′-dimethyl-4H,4′H-4,4′-bipyran were part of Group 2, being present in the individual *P. putida* treatment but not detected in the dual system.

The BMBC/Pc system exhibited an increase in the number of Group 1 compounds compared to the BMBH/Pc and BMBI/Pc systems, including L-lactic acid, 2,3-butanedione, and ethyl methoxyacetate. This suggests a more pronounced metabolic response during the interaction between *B. amyloliquefaciens* and *P. capsici*. Group 2 in BMBC/Pc included compounds such as hydrazoic acid, methylhydrazine, and 5-amino-6-nitrosopyrimidine-2,4(1H,3H)-dione, which were produced by *B. amyloliquefaciens* alone but not in the dual system, indicating possible metabolic shifts or suppression.

The BMBA/Pc system showed the greatest variation in compounds in Group 1, including acetoin, 4-penten-2-ol, 2,3-butanedione, methyl acrylate, and acetic anhydride. The increased production of these VOCs during the interaction between *B. subtilis* and *P. capsici* suggests a significant alteration in metabolic pathways, potentially linked to enhanced antimicrobial activity or signaling mechanisms.

The metabolic profiles of the individual systems revealed several compounds common between treatments, indicating their shared metabolic capabilities. *P. capsici* produced compounds such as fluoroethylene, dimethyl ether, and 4-methyl-2,4,6-tri-tert-butylcyclohexane-2,5-dien-1-one, which were also detected in bacterial isolates, suggesting overlapping biosynthetic pathways. *Bacillus* sp. and *B. subtilis* shared compounds like 4-penten-2-ol and 2-nitropropane, while *P. putida*, *B. subtilis*, and *B. amyloliquefaciens* all produced methyl tartronic acid. Additionally, *Bacillus* sp. and *P. putida* both produced isobutane, 3-methyl-1-butanol, DL-alanine, and 2-methyl-1-butanol.

Notably, the highest number of shared compounds was observed between *B. subtilis* and *B. amyloliquefaciens*, including acetoin, hydrazoic acid, 5-amino-6-nitrosopyrimidine-2,4(1H,3H)-dione, 2,3-butanedione, methyl acrylate, chloromethyl methyl ether, acetic anhydride, 1-methoxy-2-methyl-2-propanol, butylated hydroxytoluene, methylglyoxal, ketene, and ethyl methoxyacetate. Although these compounds were identified across multiple treatments, their relative abundances varied significantly, as depicted in the heatmap, highlighting differences in metabolic activity and potential functional roles.

The hierarchical clustering analysis elucidated the relationships between treatments based on their VOC profiles. Treatments clustered together exhibited similar VOC production patterns, indicating potential similarities in metabolic responses or interactions. The heatmap visually demonstrated how the presence of *P. capsici* influenced the metabolic outputs of the rhizobacteria, leading to the production of specific VOCs that might be involved in microbial communication, competition, or inhibition.

These findings underscore the dynamic nature of microbial interactions and their impact on metabolite production. The alterations in VOC abundance and composition in the dual systems suggest that the rhizobacteria may modulate their metabolic pathways in response to *P. capsici*, potentially as a mechanism of antagonism or defense. Understanding these metabolic shifts provides valuable insights into the complex interplay between beneficial microbes and plant pathogens, which could inform the development of effective biocontrol strategies and enhance sustainable agricultural practices.

## 3. Discussion

*Phytophthora capsici* is a persistent and harmful pathogen in pepper crops [2]. While chemical treatments can disrupt the pathogen’s life cycle [8], their effectiveness is increasingly compromised due to the pathogen’s ability to develop resistance [5,9]. This has led to an increased interest in exploring alternative management strategies, such as the use of biological control agents like plant growth-promoting rhizobacteria (PGPR). Antibiosis is one of the mechanisms through which PGPRs inhibit the growth and development of plant pathogens [17].

In this study, exposure to *P. capsici* to *Bacillus* sp. and *Pseudomonas putida* caused noticeable morphological alterations in the mycelium, including changes in coloration and mycelial thinning. Similar effects have been documented in other fungal pathogens, such as *Colletotrichum gloeosporioides*, which showed comparable morphological changes due to the antifungal activity of secondary metabolites produced by *Trichoderma* sp. and *Bacillus subtilis* [18]. Additionally, pigmentation changes in pathogens, such as the brown discoloration observed in this study, have been reported as stress responses triggered by environmental factors [19]. A similar stress-related response may explain the morphological changes observed in *P. capsici* during its interaction with PGPRs.

Interestingly, while *Bacillus amyloliquefaciens* (BMBC/Pc) and *Bacillus subtilis* (BMBA/Pc) exhibited a significant inhibition of *P. capsici* mycelial growth by 37.11% and 49.53%, respectively, *Pseudomonas putida* (BMBI/Pc) demonstrated a more moderate inhibitory effect, reducing mycelial growth by 12.62% (one-way ANOVA, F = 206.63, df = 4, *p* ≤ 0.05). This inhibitory action aligns with the well-documented antagonistic properties of *Pseudomonas* species against plant pathogens. While this result is unusual, certain bacteria have been shown to establish beneficial associations with fungi [20,21]. However, there are no prior reports of bacteria promoting the growth of *P. capsici*. Similar phenomena have been observed with fungicides, such as mefenoxam, which at low concentrations (1 × 10^−10^ μg/mL) has been reported to stimulate the radial growth of *Pythium* isolates by up to 10% [22]. Comparable responses have been documented in *P. aphanidermatum* when exposed to compounds like ethanol, cyazofamid, and propamocarb [23]. It is possible that the volatile organic compounds (VOCs) produced by the bacteria, combined with their concentration levels, could be inducing a hormesis effect, promoting the growth of *P. capsici* colonies.

Despite the growth-promoting effects of some bacterial isolates, the inhibitory potential of bacterial secondary metabolites against *P. capsici* is well-established. For instance, *Bacillus amyloliquefaciens* IBFCBF-1 has demonstrated the inhibition of *P. capsici* with an inhibition zone of 26.0 mm [24]. Similarly, compounds produced by *B. amyloliquefaciens* (UQ154), *Bacillus velezensis* (UQ156), and *Acinetobacter* sp. (UQ202) have shown inhibitory effects of around 35% [25]. In this study, the inhibition achieved by *B. amyloliquefaciens* and *B. subtilis* was significant and could be attributed to the production of antifungal secondary metabolites, which disrupt the pathogen’s cellular processes.

Morphological changes such as hyphal branching and swelling were also observed in *P. capsici* when exposed to *B. amyloliquefaciens* and *B. subtilis*. These alterations may reflect a disruption in polar growth, a process essential for hyphal elongation, which is likely linked to disruptions in the actin cytoskeleton [26,27]. Similar alterations, such as hyphal vacuolization, were noted with *B. subtilis*, a phenomenon that has been linked to membrane permeability issues, ion leakage, and the inhibition of cellular respiration in other fungal pathogens, such as *Fusarium oxysporum*, after their exposure to bacterial VOCs [28]. The nature and abundance of bacterial metabolites likely dictate the specific effects on the fungal colony [20].

In addition to the morphological changes, *Bacillus amyloliquefaciens* and *Bacillus subtilis* significantly reduced sporangium production. Sporangium morphological alterations, such as size reduction, have been previously reported as an adaptive response to impaired mycelial growth, resulting in cellular shrinkage and collapse. These effects may arise from increased membrane permeability, which leads to the leakage of cellular contents and the absorption of harmful compounds [29]. This phenomenon has also been documented in interactions between *Pseudomonas* sp. and *P. capsici*, where the bacteria caused changes in both hyphal and reproductive structures. These changes have been attributed to the inhibition of key enzymes involved in signal transduction pathways, which regulate critical processes like motility, proliferation, and differentiation [26].

In the detached leaf assay, the exposure of *P. capsici* mycelium to *B. amyloliquefaciens* and *B. subtilis* led to a marked reduction in virulence. This reduction in virulence may be linked to a decrease in hyphal number, alterations in sporangium morphology, and decreased sporangium production, all of which are associated with reduced zoospore motility and the inability to effectively infect plant tissues [15].

The analysis of bacterial VOCs revealed qualitative and quantitative changes in the interaction between PGPRs and *P. capsici*. It is well-documented that bacteria respond to biotic stress through the production of secondary metabolites. These responses can be long-distance, mediated by VOCs, or short-distance, mediated by diffusible organic compounds (DOCs) [30]. In vitro studies have shown that DOC-producing bacteria form a barrier between the antagonist and the pathogen, occasionally accompanied by precipitate formation in the culture medium [31,32,33]. However, in this study, none of the bacterial isolates exhibited DOC-mediated antagonisms, which led us to focus on VOC-mediated interactions.

The VOCs produced by PGPRs and *P. capsici* in individual cultures were also detected in dual culture systems, but their abundance and distribution were different. Inter- and intraspecific interactions are known to modulate VOC production, leading to changes in both diversity and abundance [8,17,29]. These findings suggest that the predominant antagonistic effect observed in this study is closely linked to the VOCs produced by the bacterial isolates.

In dual culture systems where *P. capsici* growth was promoted by *P. putida* and *Bacillus* sp., certain VOCs known for their antagonistic properties were detected. For example, DL-alanine has been shown to inhibit spore germination in *Bacillus cereus* [18], and 3-methyl-1-butanol has demonstrated an antagonistic activity against several fungal pathogens, including *Botrytis cinerea*, *Colletotrichum acutatum*, *Penicillium expansum* [34], and *Phytophthora infestans* [35]. It is possible that the promotion of *P. capsici* growth in these systems may be a result of a hormetic effect caused by the specific concentration of VOCs.

Conversely, in systems where bacterial antagonism was pronounced, such as those involving *B. subtilis* and *B. amyloliquefaciens*, other VOCs with antifungal properties were detected. For instance, L-lactic acid, produced by *Lactiplantibacillus plantarum*, has been identified as a key factor in the biocontrol of *Botrytis cinerea* [36]. Acetoin, a compound produced by *B. subtilis*, has been shown to reduce infection by *Fusarium oxysporum* f. sp. *radicis-lycopersici* in tomato plants [37]. Additionally, 2,3-butanedione has demonstrated antifungal activity against several post-harvest pathogens [38]. In *B. subtilis*-treated cultures, methylhydrazine and methylglyoxal, compounds with bactericidal and osmotic stress-inducing properties, may play a crucial role in the observed antagonism against *P. capsici* [39,40].

The differential effects observed in *P. capsici* suggest that VOCs produced by PGPRs may have multiple mechanisms of action. These mechanisms likely disrupt various biological processes in the oomycete, ultimately reducing its pathogenicity and lifecycle success, thus enhancing its biocontrol.

Moreover, some successful cases of VOC application in open-field conditions have been reported. For instance, 3-pentanol and 2-butanone have been shown to induce systemic acquired resistance in cucumber plants against *Pseudomonas syringae* pv. Lachrymans [41]. Additionally, 2,3-butanediol and its stereoisomers 2R,3R-butanediol and 2R,3S-butanediol significantly reduced virus incidence and improved pepper fruit yield [42]. Other approaches include the application of VOC-producing microorganisms individually [21] or in combination with chemical inducers [43]. However, the application of VOCs in open-field conditions is still under development. This is because biological activity depends not on a single compound but on a combination of compounds [44], which complicates their application due to the low stability of VOCs, owing to their chemical nature [21]. Furthermore, when microorganisms are directly applied to the plant, the VOC profile can vary due to biotic and abiotic factors [45], which could reduce their efficacy under field conditions.

## 4. Materials and Methods

### 4.1. Microbiological Material

The bacterial isolates used in this study included *Bacillus* sp. (BMBH), *Pseudomonas putida* (BMBI), *Bacillus subtilis* (BMBA), and *Bacillus amyloliquefaciens* (BMBC), which were obtained from the strain collection of the Molecular Biology Laboratory at CIIDIR IPN Michoacán. The *Phytophthora capsici* isolate (6143) was provided by Dr. S. P. Fernández-Pavía from the Universidad Michoacana de San Nicolás de Hidalgo. All bacterial isolates were routinely cultured on potato dextrose agar (PDA) for 48 h, while *P.capsici* was grown on a V8 agar medium for 7 days. The V8 medium was prepared by dissolving 200 mL/L of V8 juice (V8 Vegetable Juice, Campbell Soup Company, Camden, NJ, USA), 3 g/L of calcium carbonate (CaCO_3_, Sigma-Aldrich, St. Louis, MO, USA), and 16 g/L of bacteriological agar (Difco Bacto, BD, Franklin Lakes, NJ, USA) in distilled water. The mixture was autoclaved before use. Cultures were maintained under dark conditions at 27 ± 2 °C.

### 4.2. In Vitro Antagonistic Activity Assays

#### 4.2.1. Dual Culture Assays

The in vitro antimicrobial activity was evaluated using a dual culture system [46]. A V8 medium disk (8 mm diameter) containing four-day-old *P. capsici* mycelium was placed at the center of each Petri dish containing 15 mL of the PDA medium. Bacterial cultures, which had been incubated for 48 h, were streaked in four perpendicular lines at a distance of 2.5 cm from the *P. capsici* disk. The following experimental treatments were applied: *Bacillus* sp. + *P. capsici* (BMBH/Pc), *P. putida* + *P. capsici* (BMBI/Pc), *B. subtilis* + *P. capsici* (BMBA/Pc), *B. amyloliquefaciens* + *P. capsici* (BMBC/Pc), and *P. capsici* alone as a control. All Petri dishes were incubated in the dark at 27 ± 2 °C for four days. The percentage of mycelial growth inhibition was calculated after four days using the following formula:Inhibition (%) = (D − d)/D × 100(1)
where D is the diameter of the mycelium in the control (cm), and d is the diameter of the mycelium in the treated samples (cm). The experiment was designed as a completely randomized trial, with eight replicates per treatment, and was independently repeated four times.

#### 4.2.2. Morphological Alterations in *P. capsici* Mycelium

After the in vitro confrontation, morphological changes in the *P. capsici* mycelium were evaluated in the different treatments. Four days after confrontation, mycelial prints were taken from the pathogenic colony, stained with cotton blue [18], and microscopically examined using a light microscope (Carl Zeiss, Primo Star ZEISS, equipped with an Axiocam Erc 5s camera, Carl Zeiss Microscopy GmbH, Jena, Germany) at 10× and 40× magnifications. The samples were mounted on glass slides with a drop of sterile water and covered with a coverslip to observe alterations in the hyphal structure, such as thinning, branching, and vacuolization. In addition to the morphological analysis, sporangium morphology and quantity were assessed by inducing sporangia in water under fluorescent light for 48 h, followed by quantification under a stereomicroscope. The in vitro pathogenicity of *P. capsici* was also determined.

### 4.3. Morphometric and Pathogenicity Analysis

#### 4.3.1. Sporangium Induction and Quantification

Sporangium induction was performed by placing an 8 mm mycelial disk from the center of the *P. capsici* colony into a Petri dish containing 20 mL of sterile distilled water. The dishes were incubated at room temperature under fluorescent light for 48 h [15]. Sporangia were quantified under a stereomicroscope (Carl Zeiss, Stemi 305), with 18 replicates per treatment.

#### 4.3.2. Microscopic Morphometric Analysis

Morphometric analysis of the *P. capsici* mycelium was conducted using a light microscope (Carl Zeiss, Primo Star ZEISS, equipped with an Axiocam Erc 5s camera). Observations were made in four fields of view (top, bottom, left, and right) for each sample, and the images were analyzed using AxioVision software (Rel. 4.8) [15].

#### 4.3.3. Detached Leaf Assay for Pathogenicity

Leaves from one-month-old *C. annuum* cv. California Wonder plants were used. The leaves were placed in humid chambers (Petri dishes containing sterile, moistened wipes) under aseptic conditions. A 5 mm mycelial disk from each treatment was placed in the center of the leaf. A negative control, consisting of only PDA disks, was also included. The Petri dishes were incubated at room temperature in darkness. Disease severity was evaluated at 2, 4, 6, and 8 days post-inoculation (dpi) using a scale from 0 to 5, where 0 represents no damage, 1 represents a lesion affecting 1–10% of the total leaf area (TLA), 2 represents 11–20% of the TLA, 3 represents 21–30% of the TLA, 4 represents 31–50% of the TLA, and 5 indicates damage exceeding 50% of the TLA [15].

### 4.4. Qualitative and Quantitative Analysis of Volatile Organic Compounds (VOCs)

#### 4.4.1. VOC Collection from Bacterial and Dual Cultures

The production of volatile organic compounds (VOCs) was evaluated in individual bacterial isolates and in dual cultures with *P. capsici*. The bacterial isolates analyzed were *Bacillus* sp. (BMBH), *Pseudomonas putida* (BMBI), *Bacillus subtilis* (BMBA), and *Bacillus amyloliquefaciens* (BMBC). For dual cultures, the combinations BMBH/*P. capsici* (BMBH/Pc), BMBI/*P. capsici* (BMBI/Pc), BMBA/*P. capsici* (BMBA/Pc), BMBC/*P. capsici* (BMBC/Pc), and a control of *P. capsici* alone (Pc) were studied. Each treatment was performed in triplicate. The bacterial isolates were inoculated onto Petri dishes containing the PDA medium (Potato Dextrose Agar) using equidistant streaks, following the previously described protocol. In the center of each plate, a 4-day-old mycelial disk of *P. capsici* was placed, and the plates were hermetically sealed with Parafilm^®^ to prevent the loss of volatiles. Incubation was maintained at 20 ± 5 °C with 75 ± 5% relative humidity for four days.

#### 4.4.2. VOCs Extraction Using HS-SPME

VOCs extraction was performed using headspace solid-phase microextraction (HS-SPME). A 50/30 μm DVB/CAR/PDMS fiber (Divinylbenzene/Carboxen/Polydimethylsiloxane, Supelco, Bellefonte, PA, USA) was employed. Prior to each analysis, the fiber was conditioned in the GC/MS injector at 230 ± 1 °C for 15 min to eliminate potential contaminants. For the VOC collection, the Petri dishes were maintained at 25 ± 5 °C for an equilibration period of 60 min. Subsequently, the HS-SPME fiber was exposed to the headspace of the plates for 30 min to adsorb the VOCs. After exposure, the fiber was removed and immediately transferred to the gas chromatograph injector for analysis.

#### 4.4.3. VOC Analysis Using GC/MS

The chromatographic analysis was conducted using a gas chromatograph (Clarus 680, Perkin-Elmer Inc., Waltham, MA, USA) coupled to a mass spectrometer (Clarus SQ8T, Perkin-Elmer Inc., MA, USA). A 5% diphenyl/95% dimethylpolysiloxane capillary column (30 m in length, 0.32 mm in internal diameter, and 0.25 μm in film thickness) was utilized. The oven temperature program began at 40 °C for 5 min, increased to 250 °C at a rate of 9 °C/min, and was held at this temperature for an additional 5 min. Helium was used as the carrier gas at a constant flow rate of 1 mL/min. The mass spectrometer operated in electron impact mode at 70 eV, with full scan acquisition in the m/z range of 30–400. The transfer line and ionization source temperatures were set at 230 °C and 250 °C, respectively.

#### 4.4.4. Identification and Quantification of VOCs

Compound identification was performed by comparing the obtained mass spectra with those in the NIST/EPA/NIH Mass Spectral Library (version 2.2, 2014) [47]. The specific retention times of the compounds were confirmed by comparing them with available authentic standards, including eicosane (99%), ethyl acetate (≥99.5%), acetoin (≥95%), and acetic anhydride (99.5%), all purchased from Sigma Aldrich (St. Louis, MO, USA). For VOCs quantification, these authentic standards were used to construct calibration curves. Standard solutions of five different concentrations were prepared for each compound, covering the expected concentration range in the samples. Calibration curves were generated by plotting peak areas against the known concentrations of the standards. The resulting equations were used to calculate the concentrations of the compounds in the analyzed samples.

For compounds without available authentic standards, the internal standard method was employed using 1-nonanol (≥98.0%, Sigma Aldrich) as a reference. The peak areas of these compounds were normalized to the peak area of the internal standard, and the results were expressed in units of concentration equivalent to the internal standard.

#### 4.4.5. Validation of the Analytical Method

Recovery percentages of the authentic standards were evaluated to ensure the reliability of the analytical method. The obtained recovery percentages were as follows: eicosane (91–96%), ethyl acetate (75–83%), acetoin (89–93%), and acetic anhydride (71–75%). These values indicate adequate recovery and the acceptable efficiency of the extraction and analysis method. The limits of detection (LOD) for each compound were determined based on the minimum concentration that produced a signal with a signal-to-noise ratio of 3:1. The LODs obtained ranged between 0.01 and 0.09 ng/g, demonstrating the high sensitivity of the employed method. Each sample was analyzed in triplicate to ensure the precision and reproducibility of the results, maintaining variations within acceptable limits (<±25%). Blank analyses and quality controls were included to ensure the integrity of the obtained data.

### 4.5. Statistical Analysis

The data corresponding to the in vitro antagonism, sporangia quantification, and disease severity were subjected to statistical evaluation using one-way analysis of variance (ANOVA). Subsequently, Tukey’s Honestly Significant Difference (HSD) post hoc test was employed to identify significant differences between treatment groups, establishing a significance threshold of *p* ≤ 0.05. All statistical analyses were conducted using R (version 4.3.2) in conjunction with RStudio (version 2023.09.0). To investigate differences in volatile organic compound (VOC) profiles, a heatmap was constructed using the VOC matrix of each treatment. The data used for the heatmap were automatically normalized and scaled to ensure comparability between treatments. Hierarchical clustering of the treatments was performed using the Ward algorithm. Hierarchical clustering heatmaps were generated to illustrate clustering patterns, establishing a branch significance at *p* ≤ 0.05.

## 5. Conclusions

This study demonstrates that specific rhizobacterial isolates, notably *Bacillus subtilis* and *Bacillus amyloliquefaciens*, exhibit significant antagonistic effects against *Phytophthora capsici* in vitro. These effects were evidenced by the inhibition of mycelial growth and sporangia production, as well as the morphological alterations in the pathogen’s hyphae and sporangia. The analysis of the volatile organic compounds (VOCs) using GC-MS coupled with HS-SPME revealed distinctive profiles associated with these antagonistic activities. Compounds such as acetoin, L-lactic acid, and 2,3-butanedione were abundant in treatments where a significant pathogen inhibition was observed.

The findings suggest that the antagonistic effects are closely linked to the production of specific VOCs by the rhizobacteria, which may interfere with the cellular processes and signaling pathways of *P. capsici*. The modulation of VOC production during the rhizobacteria–*P. capsici* interaction indicates a dynamic metabolic response, potentially exploitable for the development of more effective biocontrol strategies.

This biocontrol approach offers a promising alternative to the use of chemical fungicides, potentially reducing their environmental impact and contributing to sustainable agricultural practices. However, to translate these results into practical applications, further research is needed to elucidate the mechanisms of action of these VOCs in plants and to evaluate the efficacy of these rhizobacteria under field conditions.

The use of *Bacillus subtilis* and *Bacillus amyloliquefaciens* as biocontrol agents represents a viable and sustainable strategy for managing *P. capsici* in horticultural crops. This approach offers an effective alternative to conventional fungicides and contributes to the development of more environmentally friendly agricultural practices.

## Figures and Tables

**Figure 1 plants-13-03224-f001:**
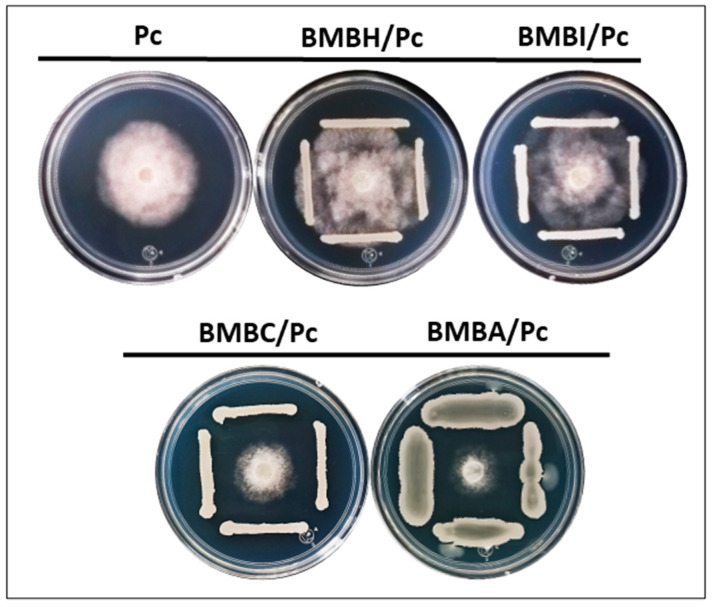
Effect of bacterial isolates on the growth and development of *Phytophthora capsici*. The dual confrontation system comprised the following treatments: *Bacillus* sp. + *P. capsici* (BMBH/Pc), *Pseudomonas putida* + *P. capsici* (BMBI/Pc), *Bacillus subtilis* + *P. capsici* (BMBA/Pc), *Bacillus amyloliquefaciens* + *P. capsici* (BMBC/Pc), and *P. capsici* alone (control).

**Figure 2 plants-13-03224-f002:**
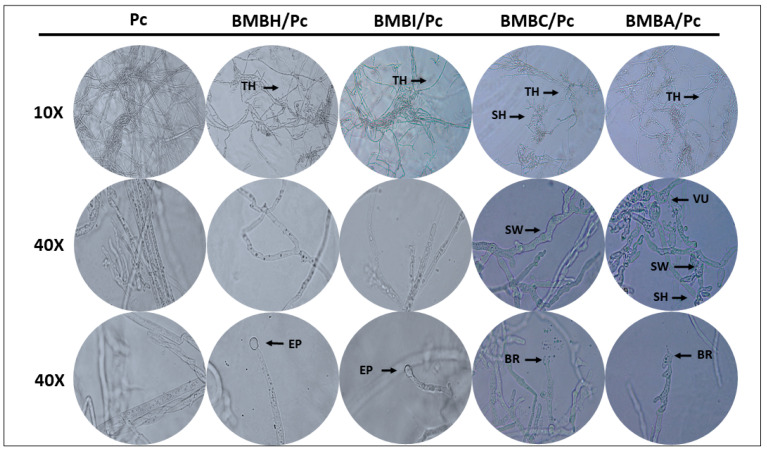
Microscopic visualization of *Phytophthora capsici* hyphal development following bacterial confrontation. Images taken at 10× and 40× magnification display mycelium from the following treatments: *Bacillus* sp. + *P. capsici* (BMBH/Pc), *Pseudomonas putida* + *P. capsici* (BMBI/Pc), *Bacillus subtilis* + *P. capsici* (BMBA/Pc), *Bacillus amyloliquefaciens* + *P. capsici* (BMBC/Pc), and *P. capsici* alone (control). Abbreviations denote observed hyphal alterations: thinning (TH), shortening (SH), sporangia formation (EP), hyphal breakage (BR), and vacuolization (VU).

**Figure 3 plants-13-03224-f003:**
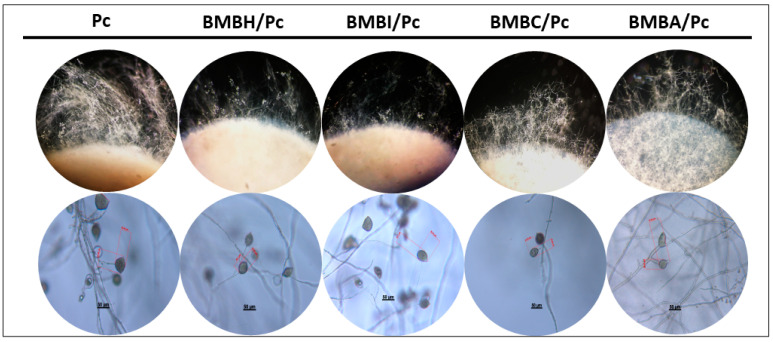
Effects of bacterial isolates on the development of *Phytophthora capsici* sporangia. The following morphological alterations of sporangia in dual confrontation systems are shown: *Bacillus* sp. + *P. capsici* (BMBH/Pc), *Pseudomonas putida* + *P. capsici* (BMBI/Pc), *Bacillus subtilis* + *P. capsici* (BMBA/Pc), *Bacillus amyloliquefaciens* + *P. capsici* (BMBC/Pc), and *P. capsici* alone (control).

**Figure 4 plants-13-03224-f004:**
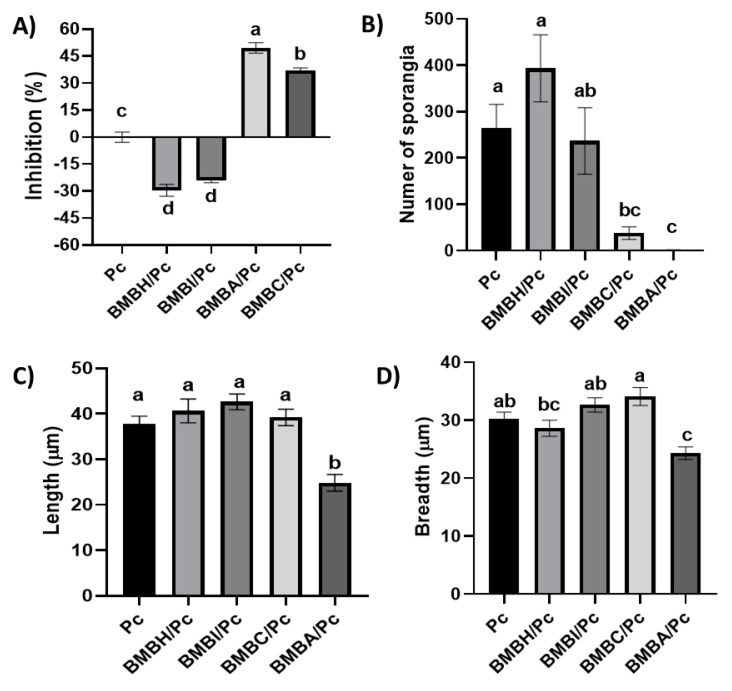
Quantitative analysis of the effects of bacterial isolates on *Phytophthora capsici*. Error bars represent the mean ± standard error. Treatments sharing the same letter are not significantly different according to Tukey’s post hoc test (*p* ≤ 0.05). (**A**) Percentage inhibition of *P. capsici* mycelial growth by bacterial treatments. Significant differences among treatments were observed (one-way ANOVA, F = 206.63, df = 4, *p* ≤ 0.05). (**B**) Effect of bacterial isolates on sporangia production. Significant differences were found among treatments (one-way ANOVA, F = 10.19, df = 4, *p* ≤ 0.05). (**C**) Effects of bacterial isolates on sporangial morphology, length (µm). A one-way ANOVA revealed significant differences (F = 13.08, df = 4, *p* ≤ 0.05). (**D**) Effects of bacterial isolates on sporangial morphology, Breadth (µm). Significant differences were detected (one-way ANOVA, F = 262.59, df = 4, *p* ≤ 0.05). Treatments: *Bacillus* sp. + *P. capsici* (BMBH/Pc), *Pseudomonas putida* + *P. capsici* (BMBI/Pc), *Bacillus subtilis* + *P. capsici* (BMBA/Pc), *Bacillus amyloliquefaciens* + *P. capsici* (BMBC/Pc), and *P. capsici* alone (control).

**Figure 5 plants-13-03224-f005:**
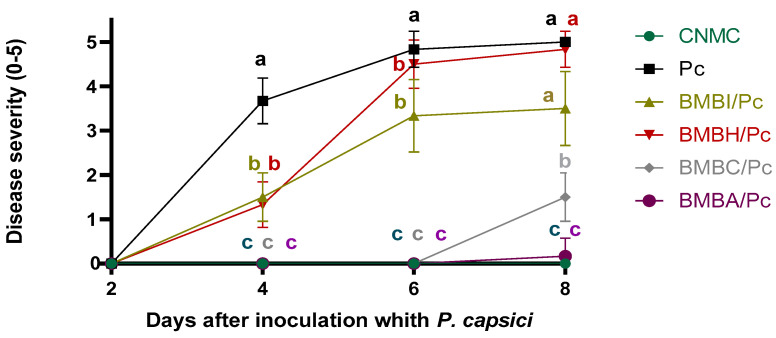
Disease severity of *Phytophthora capsici* on detached *Capsicum annuum* cv. California wonder leaves following exposure to bacterial isolates. Error bars represent the mean ± standard error. Treatments sharing the same letter are not significantly different according to Tukey’s post hoc test (*p* ≤ 0.05). Significant differences among treatments were observed on each day (one-way ANOVA: day 2, F = 90.12, df = 5, *p* ≤ 0.05; day 4, F = 177.76, df = 5, *p* ≤ 0.05; day 6, F = 243.76, df = 5, *p* ≤ 0.05; day 8, F = 243.76, df = 5, *p* ≤ 0.05). Disease severity was assessed at 2, 4, 6, and 8 days post-inoculation using a 0–5 scale: 0 = no damage; 1 = 1–10% of leaf area affected; 2 = 11–20%; 3 = 21–30%; 4 = 31–50%; 5 = more than 50% of the total leaf area. Treatments: *Bacillus* sp. + *P. capsici* (BMBH/Pc), *Pseudomonas putida* + *P. capsici* (BMBI/Pc), *Bacillus subtilis* + *P. capsici* (BMBA/Pc), *Bacillus amyloliquefaciens* + *P. capsici* (BMBC/Pc), negative control (CNMC), and *P. capsici* alone (Pc).

**Figure 6 plants-13-03224-f006:**
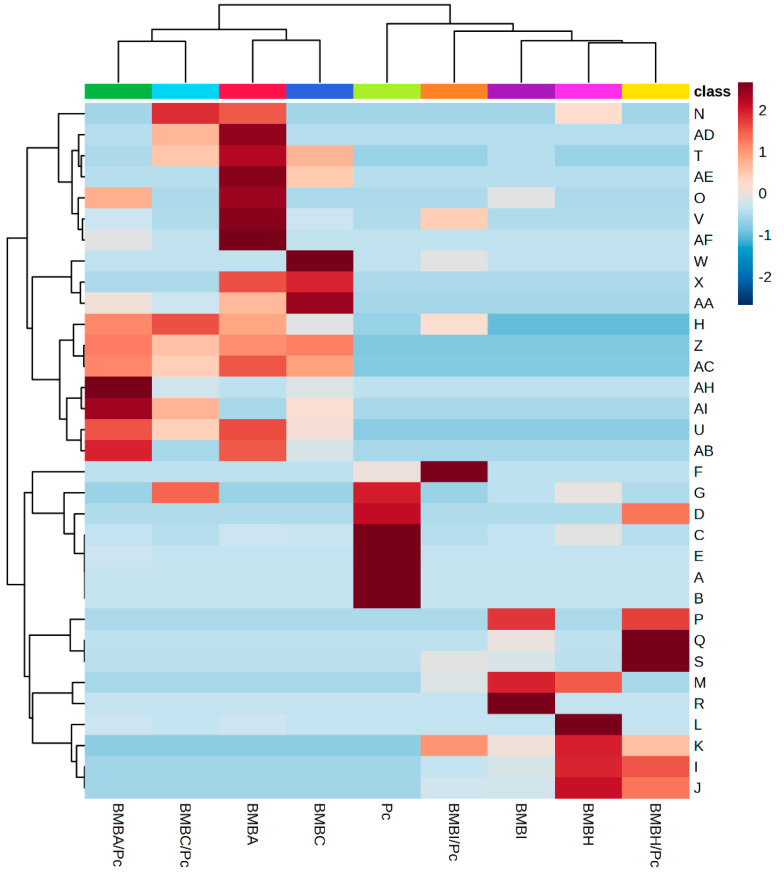
Clustering of volatile organic compounds (VOCs) produced in dual and individual systems. Each column represents an individual treatment: *Bacillus* sp. (BMBH), *Pseudomonas putida* (BMBI), *Bacillus subtilis* (BMBA), *Bacillus amyloliquefaciens* (BMBC), and dual systems with *P. capsici* (BMBH/Pc, BMBI/Pc, BMBA/Pc, BMBC/Pc), and *P. capsici* alone (control). The heatmap shows VOC abundance, with red indicating high abundance and blue indicating low abundance. Identified compounds include the following: 2-Nitropropane (N), 1-methoxy-2-methyl-2-propanol (AD), L-lactic acid (T), butylated hydroxytoluene (AE), methyltartronic acid (O), hydrazoic acid (V), methylglyoxal (AF), methylhydrazine (W), 5-amino-6-nitrosopyrimidine-2,4(1h,3h)-dione (X), methyl acrylate (AA), 4-methyl-2,4,6-tri-tert-butylcyclohexane-2,5-dien-1-one (H), 2,3-butanedione (Z), acetic anhydride (AC), ketene (AH), ethyl methoxyacetate (AI), acetoin (U), chloromethyl methyl ether (AB), ethyl acetate (F), dimethyl ether (G), eicosane (D), fluoroethylene (C), cyclopropyl carbinol (E), 1-octanamine, N-methyl-(A), 2-pentanamine (B), 2,2′,6,6′-tetra-tert-butyl-4,4′-dimethyl-4H,4′H-4,4′-bipyran (P), 1-pentanal (Q), 2-nitroethanol (S), 2-methyl-1-butanol (M), ethylenimine (R), 4-penten-2-ol (L), DL-alanine (K), isobutane (I), and 3-methyl-1-butanol (J).

**Table 1 plants-13-03224-t001:** Volatile organic compounds identified in the individual systems.

			Relative Abundance (%)				
#	RT	Compound	Pc	BMBH	BMBA	BMBC	BMBI
1	3.2	Isobutane	nd	29.98 ± 1.20	nd	nd	9.18 ± 0.37
2	4.5	Dimethyl ether	22.07 ± 0.66	1.96 ± 0.06	nd	nd	4.14 ± 0.17
3	5.0	Fluoroethylene	57.84 ± 2.31	1.45 ± 0.04	0.37 ± 0.01	1.45 ± 0.06	nd
4	6.2	2-Pentanamine	3.49 ± 0.12	nd	nd	nd	nd
5	6.8	N-Methyl-1-octanamine	3.51 ± 0.14	nd	nd	nd	nd
6	7.5	Ethyl acetate	0.44 ± 0.02	nd	nd	nd	nd
7	8.0	Cyclopropyl carbinol	9.23 ± 0.28	nd	nd	nd	nd
8	10.5	2-Nitropropane	nd	0.27 ± 0.01	0.18 ± 0.01	nd	nd
9	12.0	4-Methyl-2,4,6-tri-tert-butylcyclohexane-2,5-dien-1-one	0.36 ± 0.01	nd	0.17 ± 0.01	0.58 ± 0.02	nd
10	13.2	3-Methyl-1-butanol	nd	5.20 ± 0.21	nd	nd	6.45 ± 0.26
11	13.5	2-Methyl-1-butanol	nd	26.80 ± 1.07	nd	nd	5.64 ± 0.23
12	14.0	4-Penten-2-ol	nd	28.03 ± 1.12	0.60 ± 0.02	nd	nd
13	15.5	DL-Alanine	nd	4.33 ± 0.17	nd	nd	6.74 ± 0.27
14	16.8	1-Methoxy-2-methyl-2-propanol	nd	nd	16.29 ± 0.65	nd	nd
15	17.2	2-Nitroethanol	nd	nd	nd	nd	5.05 ± 0.20
16	17.5	1-Pentanal	nd	nd	nd	nd	1.72 ± 0.07
17	18.0	2,3-Butanedione	nd	nd	0.28 ± 0.01	1.65 ± 0.07	nd
18	18.3	Methyl acrylate	nd	nd	1.21 ± 0.05	3.04 ± 0.12	nd
19	19.0	Methylglyoxal	nd	nd	12.29 ± 0.49	nd	nd
20	20.5	Acetoin	nd	nd	25.91 ± 1.04	35.51 ± 1.42	nd
21	21.0	L-Lactic acid	nd	nd	16.42 ± 0.66	13.18 ± 0.53	25.31 ± 1.01
22	22.0	Methyltartronic acid	nd	nd	21.64 ± 0.87	nd	32.70 ± 1.31
23	23.5	Hydrazoic acid	nd	nd	1.09 ± 0.04	0.68 ± 0.03	nd
24	24.0	Methylhydrazine	nd	nd	nd	6.03 ± 0.24	nd
25	25.0	5-Amino-6-nitrosopyrimidine-2,4(1H,3H)-dione	nd	nd	0.74 ± 0.03	3.12 ± 0.12	nd
26	26.5	Chloromethyl methyl ether	nd	nd	0.99 ± 0.04	9.61 ± 0.38	nd
27	27.0	Acetic anhydride	nd	nd	1.39 ± 0.06	4.44 ± 0.18	nd
28	27.5	Ketene	nd	nd	nd	0.29 ± 0.01	nd
29	28.0	Butylated hydroxytoluene	nd	nd	0.35 ± 0.01	0.45 ± 0.02	nd
30	29.0	2,2′,6,6′-Tetra-tert-butyl-4,4′-dimethyl-4H,4′H-4,4′-bipyran	nd	nd	nd	nd	2.43 ± 0.10
31	30.0	Eicosane	3.06 ± 0.15	nd	nd	nd	nd
32	31.5	Ethyl methoxyacetate	nd	nd	nd	19.14 ± 0.77	Nd

The means ± standard deviations are presented. # indicates the compound number for organizational purposes. nd indicates not detected. RT: Retention time (minutes). Relative abundances are expressed as percentages (%). Treatments: Pc (*Phytophthora capsici* alone), BMBH (*Bacillus* sp.), BMBA (*Bacillus subtilis*), BMBC (*Bacillus amyloliquefaciens*), BMBI (*Pseudomonas putida*). Compounds were identified using gas chromatography-mass spectrometry (GC-MS) coupled with headspace solid-phase microextraction (HS-SPME). The RT values correspond to data reported previously in the literature or in the NIST database (http://www.nist.gov).

**Table 2 plants-13-03224-t002:** Volatile organic compounds identified in the dual systems.

			Relative Abundance (%)			
#	RT	Compound	BMBA/Pc	BMBC/Pc	BMBH/Pc	BMBI/Pc
1	3.2	Isobutane	nd	nd	28.04 ± 1.05	16.25 ± 0.65
2	4.5	Dimethyl ether	nd	2.78 ± 0.11	2.33 ± 0.08	nd
3	5.0	Fluoroethylene	0.18 ± 0.007	nd	nd	nd
4	6.2	2-Pentanamine	nd	nd	nd	nd
5	6.8	N-Methyl-1-octanamine	nd	nd	nd	nd
6	7.5	Ethyl acetate	nd	nd	nd	13.42 ± 0.55
7	8.0	Cyclopropyl carbinol	0.06 ± 0.002	nd	nd	nd
8	10.5	2-Nitropropane	nd	0.51 ± 0.02	nd	nd
9	12.0	4-Methyl-2,4,6-tri-tert-butylcyclohexane-2,5-dien-1-one	0.08 ± 0.003	0.28 ± 0.012	nd	5.48 ± 0.22
10	13.2	3-Methyl-1-butanol	nd	nd	17.39 ± 0.70	6.19 ± 0.25
11	13.5	2-Methyl-1-butanol	nd	nd	nd	6.56 ± 0.27
12	14.0	4-Penten-2-ol	1.07 ± 0.04	nd	nd	nd
13	15.5	DL-Alanine	nd	nd	9.57 ± 0.38	4.10 ± 0.17
14	16.8	1-Methoxy-2-methyl-2-propanol	nd	17.45 ± 0.70	nd	nd
15	17.2	2-Nitroethanol	nd	nd	27.88 ± 1.10	19.38 ± 0.80
16	17.5	1-Pentanal	nd	nd	8.08 ± 0.30	nd
17	18.0	2,3-Butanedione	1.20 ± 0.05	3.01 ± 0.12	nd	nd
18	18.3	Methyl acrylate	1.97 ± 0.08	1.17 ± 0.05	nd	nd
19	19.0	Methylglyoxal	1.88 ± 0.08	nd	nd	nd
20	20.5	Acetoin	44.53 ± 1.80	20.06 ± 0.80	1.46 ± 0.06	nd
21	21.0	L-Lactic acid	2.51 ± 0.10	32.78 ± 1.30	nd	nd
22	22.0	Methyltartronic acid	2.13 ± 0.08	nd	nd	nd
23	23.5	Hydrazoic acid	0.26 ± 0.01	nd	nd	19.57 ± 0.80
24	24.0	Methylhydrazine	nd	nd	nd	5.41 ± 0.22
25	25.0	5-Amino-6-nitrosopyrimidine-2,4(1H,3H)-dione	nd	nd	nd	nd
26	26.5	Chloromethyl methyl ether	14.64 ± 0.60	nd	nd	nd
27	27.0	Acetic anhydride	2.03 ± 0.08	1.74 ± 0.07	nd	nd
28	27.5	Ketene	1.20 ± 0.05	0.12 ± 0.005	nd	nd
29	28.0	Butylated hydroxytoluene	nd	nd	nd	nd
30	29.0	2,2′,6,6′-Tetra-tert-butyl-4,4′-dimethyl-4H,4′H-4,4′-bipyran	nd	nd	1.76 ± 0.07	nd
31	30.0	Eicosane	nd	nd	3.47 ± 0.14	nd
32	31.5	Ethyl methoxyacetate	26.19 ± 1.00	20.06 ± 0.80	nd	nd

The means ± standard deviations are presented. # indicates the compound number for organizational purposes. nd indicates not detected. RT: Retention time (minutes). Relative abundances are expressed as percentages (%). BMBA/Pc (*Bacillus subtilis* + *P. capsici*), BMBC/Pc (*Bacillus amyloliquefaciens* + *P. capsici*), BMBH/Pc (*Bacillus* sp. + *P. capsici*), BMBI/Pc (*Pseudomonas putida* + *P. capsici*). Compounds were identified using gas chromatography-mass spectrometry (GC-MS) coupled with headspace solid-phase microextraction (HS-SPME). The RT values correspond to data reported previously in the literature or in the NIST database (http://www.nist.gov).

## Data Availability

The data presented in this study are available in this article.

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
