# Peer review of "Antagonistic Effects and Volatile Organic Compound Profiles of Rhizobacteria in the Biocontrol of *Phytophthora capsici"

_plants, 2024, doi:10.3390/plants13223224_

Round 1

Reviewer 1 Report

Comments and Suggestions for Authors

These are my main comments on the manuscript (Plants-3275353) entitled “Antagonistic Effects and Volatile Organic Compound Profiles of Rhizobacteria in the Biocontrol of Phytophthora capsici”. This work investigates the antagonistic effects of six rhizobacterial isolates against P. capsici, focusing on the production of volatile organic compounds (VOCs). However, details about introduction, materials and methods, and results section are needed. Following substantial revisions should be incorporated in the manuscript prior to acceptance.

Ls.26-27: Keywords should be in alphabetic order. Also, keywords serve to widen the opportunity to be retrieved from a database. To put words that already are into title and abstracts makes KW not useful. Please choose terms that are neither in the title nor in abstract.

L.30: Provide order and plant family taxa.

L.35: Which peppers species?

L.56: Delete “of those”

L.58: Delete “of these challenges”

L.75: A hypothesis for this study is needed.

Ls.75, 82, 351, etc.: In vitro should be in italic.

Ls.77-78: Delete “volatile organic compounds”

L.84: Change “control treatment” by “control group”. Check in all manuscript.

Ls.89, 98, 118, 141, 143: (p ≤ 0.05)? Please, provide the F-value, Degree Freedom, and P-value by each ANOVA result.

Ls.105-112: In figure 2, What is the difference between 10x, 40x and 40x?

Ls.125-135: Again, provide the F-value, Degree Freedom, and P-value by each ANOVA result.

Ls.136-153: Graphic is confusing, change by column bars. Also, provide the F-value, Degree Freedom, and P-value by each ANOVA result.

Table 1 and 2: Explain “#” and “nd”.

Ls.312-314 and 366-368: Hypothesis should be in introduction section.

Ls.391-392: … consisting of 200 mL V8 juice, 3 g CaCO₃, and 16 g bacteriological…

L.392: For cultures, Any time-period?

L.409: P. capsica should be in italic.

Ls.412-413: How were they examined microscopically? Explain.

Ls.428-429: Delete this sentence.

Ls.450-453: Combine and summarize these sentences. Delete “VOCs were analyzed using gas chromatography coupled with mass spectrometry 452

(GC/MS).”.

Reviewer 2 Report

Comments and Suggestions for Authors

In this study, the authors investigated the antagonistic effects of rhizobacterial isolates against the phytopathogen Phytophthora capsici. They demonstrated that rhizobacteria, specifically two isolates, Bacillus subtilis and Bacillus amyloliquefaciens, possess antifungal properties, significantly reducing the pathogen's growth under in vitro conditions. A detached leaf assay further showed that P. capsici exhibited reduced virulence after interaction with B. subtilis or B. amyloliquefaciens. This study provides important findings and provides valuable insights into the potential use of rhizobacterial isolates as a biocontrol agent against P. capsica. In my opinion, the main limitation of this study is that the authors evaluated only the effect of VOCs not the diffusible compounds. Also, the authors did not confirm the effect of VOCs on fungal growth or pathogenicity.

Major comments:

1.     The authors did not evaluate the production and effects of diffusible compounds, which could have provided a more comprehensive understanding of the bacterial-fungal interaction. The authors mentioned in Lines 351-352 that precipitation may occur if diffusible compounds are involved. But the formation of precipitation it is not guaranteed. Sometimes the precipitation may not be visible depends on the pH or ingredients in the medium used. To confirm the antagonistic effects of rhizobacterial VOCs on P. capsici, an in vitro assay should be conducted using a petri dish system that allows the bacteria and fungus to grow in the same environment without any direct contact. This setup would confirm that only VOCs influence fungal growth, eliminating the potential effects of diffusible compounds.

2.     Figure 2: The authors indicate the Phytophthora capsica hyphae looks thinner in the treatment than the control. It’s hard to see the difference in the 10X. In 40X, the hyphae don’t appear thinner in the treatment compared to the control especially for BMBH/Pc, BMBI/Pc and BMBC/Pc. 

3.     In Lines 95-98, authors mentioned that P. putida reduced the pathogen growth by 12.62% (p<0.05). However, in the discussion section, (Lines 304-305) the authors mentioned that P. putida promoted mycelial growth by 24.28%. The image and Figure 4A show that pathogen growth is not reduced. Please clarify.

Minor comments:

Line 18: Five rhizobacterial isolates not six. The authors listed in the abstract and throughout the manuscript on only five rhizobacterial isolates.

Figure 1: Labelling error. Paenibacillus sp + P. capsica is labelled as BMPA/Pc in the figure but given as BMBP/Pc in the legend. 

Figure 2 and Figure 3: Please ensure all abbreviations used in the figure are listed in the legend. Also, the legend indicates that the BMBP/Pc is given in the figure, but BMBP/Pc data is missing in the figure.

Round 2

Reviewer 1 Report

Comments and Suggestions for Authors

While I have no objection against publishing the data, I have a only issue that need addressing. The authors did not clarify the statistical values (F-value, degree freedom, and P-value)  to explain each result obtained in the experiments as suggested in the anterior review. I am concerned about the poorly ANOVA elements in the results. The authors should provide more information on the statistical values to make this version acceptable for publication.

Reviewer 2 Report

Comments and Suggestions for Authors

The authors have answered all the comments raised in my review, and the manuscript is much improved compared to the previous version. I am satisfied with their responses.

Round 3

Reviewer 1 Report

Comments and Suggestions for Authors

The manuscript “Antagonistic Effects and Volatile Organic Compound Profiles of Rhizobacteria in the Biocontrol of Phytophthora capsici” has been improved and all my questions were taken into account. I recommend the publication in “Plants”. 

Author Response

Dear Reviewer,

Thank you very much for your positive evaluation of our manuscript titled "Antagonistic Effects and Volatile Organic Compound Profiles of Rhizobacteria in the Biocontrol of Phytophthora capsici". We are pleased to hear that our revisions have adequately addressed all your questions. We appreciate your recommendation for publication in "Plants".